# Surface to bulk Fermi arcs via Weyl nodes as topological defects

Kun Woo Kim[1], Woo-Ram Lee[1,2], Yong Baek Kim[1,3] & Kwon Park[1,2]

A hallmark of Weyl semimetal is the existence of surface Fermi arcs. An intriguing question is what determines the connectivity of surface Fermi arcs, when multiple pairs of Weyl nodes are present. To answer this question, we show that the locations of surface Fermi arcs are predominantly determined by the condition that the Zak phase integrated along the normal-to-surface direction is $\pi$. The Zak phase can reveal the peculiar topological structure of Weyl semimetal directly in the bulk. Here, we show that the winding of the Zak phase around each projected Weyl node manifests itself as a topological defect of the Wannier–Stark ladder, energy eigenstates under an electric field. Remarkably, this leads to bulk Fermi arcs, open-line segments in the bulk spectra. Bulk Fermi arcs should exist in conjunction with surface counterparts to conserve the Weyl fermion number under an electric field, which is supported by explicit numerical evidence.

[1] School of Physics, Korea Institute for Advanced Study, Seoul 02455, Korea. [2] Quantum Universe Center, Korea Institute for Advanced Study, Seoul 02455, Korea. [3] Department of Physics and Centre for Quantum Materials, University of Toronto, Toronto, Ontario, Canada M5S 1A7. Correspondence and requests for materials should be addressed to K.P. (email: kpark@kias.re.kr).

Weyl semimetal is a gapless, topological phase of matter, which can be generated quite generally by breaking either time-reversal or inversion symmetry near the phase boundary between topological and trivial insulators. One of the most dramatic properties of Weyl semimetal is that surface states have a Fermi surface consisting of open-line segments called surface Fermi arcs[1–12]. Each surface Fermi arc connects two surface-projected Weyl nodes with opposite chiralities, playing an important role in resolving the chiral anomaly of Weyl fermion[13–21].

An intriguing question is what determines the connectivity of surface Fermi arcs, when multiple pairs of Weyl nodes are present. In this work, we answer this question by showing that the locations of surface Fermi arcs are predominantly determined by the condition that the Zak phase integrated along the normal-to-surface direction is $\pi$. The Zak phase is the Berry phase integrated along a straight, but closed path in the momentum space traversing the entire one-dimensional Brillouin zone[22].

More importantly, the Zak phase can reveal the peculiar topological structure of Weyl semimetal directly in the bulk. It has been shown in a previous work[23] that the non-trivial topological order of topological insulator can be directly manifested in the winding number of the Wannier–Stark ladder (WSL), which is ultimately governed by the topological structure of the Zak phase. The WSL is the energy eigenstates of electrons confined in the lattice under an electric field. Physically speaking, the WSL can be thought as the quantized modes of the Bloch oscillation in a similar way that phonons are those of the lattice vibration.

Here, we show that, in Weyl semimetal, the Zak phase winds by $2\pi$ around each projected Weyl node, creating a screw dislocation in the energy spectrum of WSL eigenstates. Eventually, these topological defects induce open-line segments in the momentum spectra of WSL, which we call bulk Fermi arcs. We provide an argument that the existence of bulk Fermi arcs is actually required to conserve the Weyl fermion number under an electric field, which is supported by explicit numerical evidence.

## Results

**Connectivity of surface Fermi arcs**. We ask if a certain bulk property of the system can determine the connectivity of surface Fermi arcs. An answer to this question would provide valuable information to characterize Weyl semimetal without solving complicated eigenvalue equations of the microscopic Hamiltonian with an open boundary condition.

To this end, let us begin by considering graphene, which is a two-dimensional Dirac/Weyl semimetal. In graphene, edge states appear depending on the edge orientation. Delplace *et al.*[24] proposed an idea that the existence of edge states is related with the condition that the Zak phase integrated along the normal-to-edge direction is $\pi$. This can be proved rigorously for certain edge orientations, while numerically confirmed in general.

Meanwhile, Mong and Shivamoggi[25] provided a related, but somewhat more general proof for the existence condition of edge/surface states in two-/three-dimensional (3D) topological insulators. Specifically, they considered the Dirac Hamiltonian, which can be written as

$$H = \mathbf{h} \cdot \mathbf{\Gamma} = \left[ \mathbf{b}(\mathbf{k}_\perp)e^{-ik_\parallel a} + \mathbf{b}_0(\mathbf{k}_\perp) + \mathbf{b}^*(\mathbf{k}_\perp)e^{ik_\parallel a} \right] \cdot \mathbf{\Gamma}, \quad (1)$$

where $\mathbf{k}_\perp$ and $\mathbf{k}_\parallel$ ($k_\parallel = |\mathbf{k}_\parallel|$) are the momenta perpendicular and parallel to the normal direction to the edge/surface, respectively. $\mathbf{\Gamma}$ is a vector composed of the gamma matrices satisfying the Clifford algebra. $a$ is the lattice constant along $\mathbf{k}_\parallel$. An important assumption above is that hopping occurs only between nearest neighbours along the normal direction to the edge/surface. Under

this assumption, the curve traced by $\mathbf{h}$ as a function of $\mathbf{k}_\parallel$ forms an ellipse, whose semi-major and semi-minor axes are $2\mathrm{Re}[\mathbf{b}(\mathbf{k}_\perp)]$ and $2\mathrm{Im}[\mathbf{b}(\mathbf{k}_\perp)]$ with its centre located at $\mathbf{b}_0(\mathbf{k}_\perp)$. It is proved in ref. 25 that an edge/surface state exists at $\mathbf{k}_\perp$ if and only if the projection of the $\mathbf{h}$ curve onto the $\mathrm{Re}[\mathbf{b}(\mathbf{k}_\perp)] - \mathrm{Im}[\mathbf{b}(\mathbf{k}_\perp)]$ plane encloses the origin of $\mathbf{h} = 0$. Moreover, the energy of such an edge/surface state is equal to the distance between the origin and the plane containing the $\mathbf{h}$ curve. This means that zero-energy edge/surface states occur when the origin is enclosed by the $\mathbf{h}$ curve, lying within the same plane.

For two-band models, this existence condition for zero-energy edge/surface states can be nicely rephrased in terms of the Zak phase. In two-band models, where $\Gamma$ is replaced by $\sigma$, there is a Dirac monopole with monopole strength $q = \pm 1/2$ at the origin of $\mathbf{h} = 0$, generating the radial Berry curvature. Then, the above existence condition is precisely equivalent to the condition that the Berry phase integrated along the $\mathbf{h}$ curve is $\pi$, which is half the solid angle of an equator. In turn, this particular Berry phase is nothing but the Zak phase integrated along the normal direction to the edge/surface, that is, $\gamma_\alpha^{\mathrm{Zak}}(\mathbf{k}_\perp) = \oint d\mathbf{k}_\parallel \cdot \mathcal{A}_\alpha(\mathbf{k})$ with the Berry connection $\mathcal{A}_\alpha(\mathbf{k}) = \langle \phi_\alpha(\mathbf{k})|i\nabla_\mathbf{k}|\phi_\alpha(\mathbf{k})\rangle$, where $\phi_\alpha(\mathbf{k})$ is the periodic part of the Bloch wave function in the $\alpha$-th band, which can be either valence or conduction band in two-band models.

Strictly, the applicability of the above existence condition is limited to two-band models with nearest neighbour hopping. However, this limitation can be somewhat relaxed considering that, by its intrinsic nature, the microscopic Hamiltonian of every Weyl semimetal can be accurately approximated as a two-band, low-energy effective Hamiltonian, which is obtained by expanding the microscopic Hamiltonian up to second order of momenta near Weyl nodes. By performing $k \to \frac{1}{a}\sin ka$ and $k^2 \to \frac{2}{a^2}(1 - \cos ka)$, one can then construct a minimally lattice-regularized two-band Hamiltonian with nearest neighbour hopping. Provided that the connectivity of surface Fermi arcs is well captured by such a minimally lattice-regularized Hamiltonian, we predict that the locations of surface Fermi arcs (which are the zero-energy surface states) are predominantly determined by the condition that the Zak phase integrated along the normal-to-surface direction is $\pi$. This prediction is confirmed to be accurate in various theoretical models.

**Topological defects of the Wannier–Stark ladder**. The Zak phase can reveal the peculiar topological structure of Weyl semimetal directly in the bulk through the WSL emerging under an electric field. To get physical intuitions for what this means and how this is possible, it is helpful to first understand that the WSL is the quantized modes of the Bloch oscillation, which is the semiclassical motion of electrons confined in the lattice under an electric field. In this situation, electrons are accelerated, travelling through the momentum space until they hit the Brillouin zone boundary. Then, due to the periodic boundary condition, they reappear at the opposite end of the Brillouin zone, eventually forming closed orbits. Such closed orbits can be quantized upon switching from the semiclassical to quantum description. The so-obtained quantized modes are the WSL eigenstates.

The quantization procedure becomes particularly simple and intuitive under the adiabatic condition that the electric field is not too strong to cause mixing between different bands, that is, there is no Zener tunnelling. Under this condition, the Bloch oscillation originating from each band can be individually quantized via the Bohr–Sommerfeld quantization rule. Specifically, the energy of WSL eigenstates is given as follows[23]:

$$\mathcal{E}_{\alpha,n}^{\mathrm{WSL}}(\mathbf{k}_\perp) = \bar{\mathcal{E}}_\alpha(\mathbf{k}_\perp) + eaE\left[n + \frac{\gamma_\alpha^{\mathrm{Zak}}(\mathbf{k}_\perp)}{2\pi}\right], \quad (2)$$

where $\bar{\mathcal{E}}_\alpha(\mathbf{k}_\perp) = \frac{a}{2\pi} \oint dk_\parallel \mathcal{E}_\alpha(\mathbf{k})$ is the one dimensionally averaged energy of the $\alpha$-th band, $E$ is the electric-field strength and $n$ ($\in \mathbb{Z}$) is the WSL index. $\gamma_\alpha^{Zak}(\mathbf{k}_\perp)$ is same as the above Zak phase except that, here, $\mathbf{k}_\perp$ and $\mathbf{k}_\parallel$ are the momenta perpendicular and parallel to the electric field, respectively. Physically, the Zak phase indicates the shift in the mean position of the WSL eigenstate wave packets, identified as the polarization[22,26].

To demonstrate concretely how the Zak phase reveals the peculiar topological structure of Weyl semimetal, let us consider the model Hamiltonian proposed by Yang et al.[2] which describes a time-reversal symmetry-broken Weyl semimetal:

$$H(\mathbf{k}) = \left[ -2t(\cos k_x - \cos k_0) + m(2 - \cos k_y - \cos k_z) \right]\sigma_x + 2t \sin k_y \sigma_y + 2t \sin k_z \sigma_z, \tag{3}$$

which has two Weyl nodes at $\mathbf{k} = (\pm k_0, 0, 0)$. From this forward, all momenta are denoted in units of $1/a$ unless stated otherwise.

Figure 1a shows the zero-energy momentum spectrum of surface states residing in a $y$ axis-cut surface, where $k_x$ and $k_z$ are good quantum numbers. As one can see, there exists a surface Fermi arc connecting two surface-projected Weyl nodes at $(k_x, k_z) = (\pm k_0, 0)$. It was predicted in the previous section that the locations of surface Fermi arcs are predominantly determined by the condition that the Zak phase integrated along the normal-to-surface direction, which is the $y$ direction here, is $\pi$. Figure 1b shows that this prediction is accurate.

More importantly, each projected Weyl node creates a screw dislocation in the Zak phase. See Fig. 1c for the 3D plot of the

Zak phase. Such a screw dislocation in the Zak phase manifests itself as a topological defect of the WSL. Figure 1d shows the zero-energy momentum spectrum of WSL eigenstates generated from the valence band, which is obtained via the adiabatic formula in equation (2). Specifically, in Fig. 1d, we plot the following spectral function at $\omega = 0$:

$$\rho_\alpha(\omega, \mathbf{k}_\perp) = \frac{1}{\pi} \text{Im} \sum_n \left[ \frac{1}{\omega - \mathcal{E}_{\alpha,n}^{WSL}(\mathbf{k}_\perp) + i\eta} \right], \tag{4}$$

which exhibits various spectral peaks following the trajectory of $\omega = \mathcal{E}_{\alpha,n}^{WSL}(\mathbf{k}_\perp)$. Note that $\mathcal{E}_{\alpha,n}^{WSL}(\mathbf{k}_\perp)$ becomes multi-valued if there is a screw dislocation in the Zak phase. Consequently, the zero-energy momentum spectrum of WSL eigenstates can show, in addition to many closed loops, an open-line segment connecting two projected Weyl nodes with opposite chiralities similar to the surface Fermi arc. We call this open-line segment the bulk Fermi arc.

In fact, the conduction band generates a similar bulk Fermi arc (as well as other closed-loop WSL eigenstates), which, incidentally, is exactly overlapped with the valence-band counterpart at zero energy in the above model Hamiltonian. Fortunately, it turns out that the bulk Fermi arc remains robust despite mixing between WSL eigenstates generated from both valence and conduction bands. In other words, the bulk Fermi arc can persist even beyond the strictly valid regime of the adiabatic condition, that is, $eaE/t \ll 1$.

To confirm this, we compute the momentum spectrum of WSL eigenstates by directly diagonalizing the microscopic model

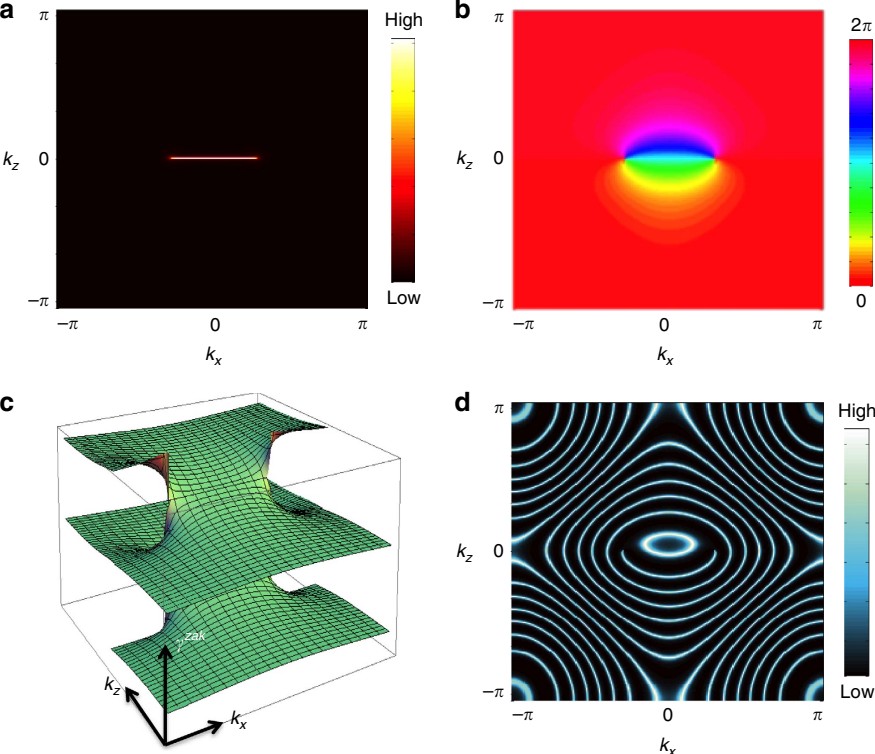

**Figure 1 | Surface and bulk Fermi arcs in a time-reversal symmetry-broken Weyl semimetal.** Here, we analyse the model Hamiltonian proposed by Yang et al. in equation (3) with model parameters chosen so that $k_0 = 2\pi/7$ and $m/t = 2$. (**a**) Zero-energy momentum spectrum of $y$ axis-cut surface states showing the trajectory of a surface Fermi arc, which is obtained by solving eigenvalue equations of the model Hamiltonian with an open boundary condition. (**b**) Zak phase of the valence band integrated along the $y$ axis. (**c**) 3D plot of the Zak phase showing that each projected Weyl node creates a screw dislocation in the Zak phase. (**d**) Zero-energy momentum spectrum of WSL eigenstates generated from the valence band, which is obtained via the adiabatic formula in equations (2) and (4). Here, the electric field is applied along the $y$ direction with its strength set equal to $eaE/t = 0.25$. Note that the momentum spectrum of WSL eigenstates is periodic in energy with period of $eaE$.

Hamiltonian under an electric field. Specifically, we compute the following spectral function, which is constructed in terms of the exact eigenstates of the microscopic Hamiltonian in the presence of electrostatic potential:

$$\rho(\omega, \mathbf{k}_\perp) = -\frac{1}{\pi}\mathrm{ImTr}\left[\frac{1}{\omega - \tilde{H}(\mathbf{k}_\perp) - V + i\eta}\right], \qquad (5)$$

where $\tilde{H}(\mathbf{k}_\perp)$ is obtained by Fourier-transforming the microscopic model Hamiltonian with respect to $k_y$; $[\tilde{H}(\mathbf{k}_\perp)]_{n_y,n_y'} = \frac{1}{2\pi}\int dk_y H(\mathbf{k})e^{ik_y a(n_y - n_y')}$, where $n_y$ is the layer index along the $y$ direction. Note that the trace Tr is taken over both $n_y$ and pseudospin index. The electrostatic potential term is given by $V = eaE(n_y - N_y/2)\delta_{n_y,n_y'}$, where the electrostatic potential is set to be zero at the middle of the system.

Figure 2 shows various zero-energy cuts of the above spectral function as a function of electric-field strength. In particular, Fig. 2a is computed at the same electric-field strength as Fig. 1d. As one can see, the two figures are essentially identical, showing that the adiabatic formula provides an excellent approximation to the exact results, at least at this range of electric-field strengths. Figure 2b–d show that the bulk Fermi arc persists up to reasonably strong electric fields.

The model Hamiltonian in equation (3) provides a convenient platform to study various topological properties of Weyl semimetal. The applicability of this model, however, is somewhat limited since it requires a breaking of the time-reversal symmetry. Another pathway to generate Weyl semimetal is to break the inversion symmetry while preserving the time-reversal symmetry, which may be more relevant in view of recent experimental confirmations of Weyl semimetal in TaAs[6–12]. In this work, we

focus on the tight-binding model Hamiltonian proposed by Ojanen[5], which describes a time-reversal invariant Weyl semimetal:

$$H(\mathbf{k}) = d_1(\mathbf{k})\sigma_x + d_2(\mathbf{k})\sigma_y + \left[\epsilon + \sum_{\alpha=x,y,z} D_\alpha(\mathbf{k})s_\alpha\right]\sigma_z, \qquad (6)$$

where $d_1(\mathbf{k}) = t\,(1 + \cos\,\mathbf{k}\cdot\mathbf{a}_1 + \cos\,\mathbf{k}\cdot\mathbf{a}_2 + \cos\,\mathbf{k}\cdot\mathbf{a}_3)$, $d_2(\mathbf{k}) = t\,(\sin\,\mathbf{k}\cdot\mathbf{a}_1 + \sin\,\mathbf{k}\cdot\mathbf{a}_2 + \sin\,\mathbf{k}\cdot\mathbf{a}_3)$ and $D_x(\mathbf{k}) = \lambda[\sin\mathbf{k}\cdot\mathbf{a}_2 - \sin\mathbf{k}\cdot\mathbf{a}_3 - \sin(\mathbf{k}\cdot\mathbf{a}_2 - \mathbf{k}\cdot\mathbf{a}_1) + \sin(\mathbf{k}\cdot\mathbf{a}_3 - \mathbf{k}\cdot\mathbf{a}_1)]$ with $\mathbf{a}_1 = \frac{a}{2}(0,\ 1,\ 1)$, $\mathbf{a}_2 = \frac{a}{2}(1,\ 0,\ 1)$, and $\mathbf{a}_3 = \frac{a}{2}(1,\ 1,\ 0)$. (Here, we reintroduce the lattice constant $a$ for clarity.) $(\sigma_x, \sigma_y, \sigma_z)$ and $(s_x, s_y, s_z)$ are the Pauli matrices acting on the sublattice and spin basis, respectively. The other components, $D_y(\mathbf{k})$ and $D_z(\mathbf{k})$, are obtained by permuting $\mathbf{a}_i$ ($i = 1, 2, 3$) cyclically from the expression of $D_x(\mathbf{k})$.

The above Hamiltonian has four bands composed of two conduction and two valence bands, among which the middle two bands, that is, the top valence and bottom conduction bands constitute a Weyl semimetal. Concretely, the Hamiltonian can be decomposed conveniently into two block-diagonalized sublattice-basis Hamiltonians, $H_\sigma^\pm$, by first diagonalizing the spin-basis part of the Hamiltonian, $H_s = \sum_\alpha D_\alpha s_\alpha$:

$$H_\sigma^\pm(\mathbf{k}) = d_1(\mathbf{k})\sigma_x + d_2(\mathbf{k})\sigma_y + [\epsilon \pm D(\mathbf{k})]\sigma_z, \qquad (7)$$

where $D(\mathbf{k}) = \sqrt{\sum_\alpha D_\alpha^2(\mathbf{k})}$. For $\epsilon > 0$ ($\epsilon < 0$), $H_\sigma^-$ ($H_\sigma^+$) describes a Weyl semimetal composed of the middle two bands, provided that $|\epsilon| < 4|\lambda|$, in which case there are 12 inequivalent Weyl nodes in the first Brillouin zone. These Weyl nodes are located at $(\pm k_0, 0, 2\pi)$, $(0, \pm k_0, 2\pi)$, $(\pm k_0, 2\pi, 0)$,

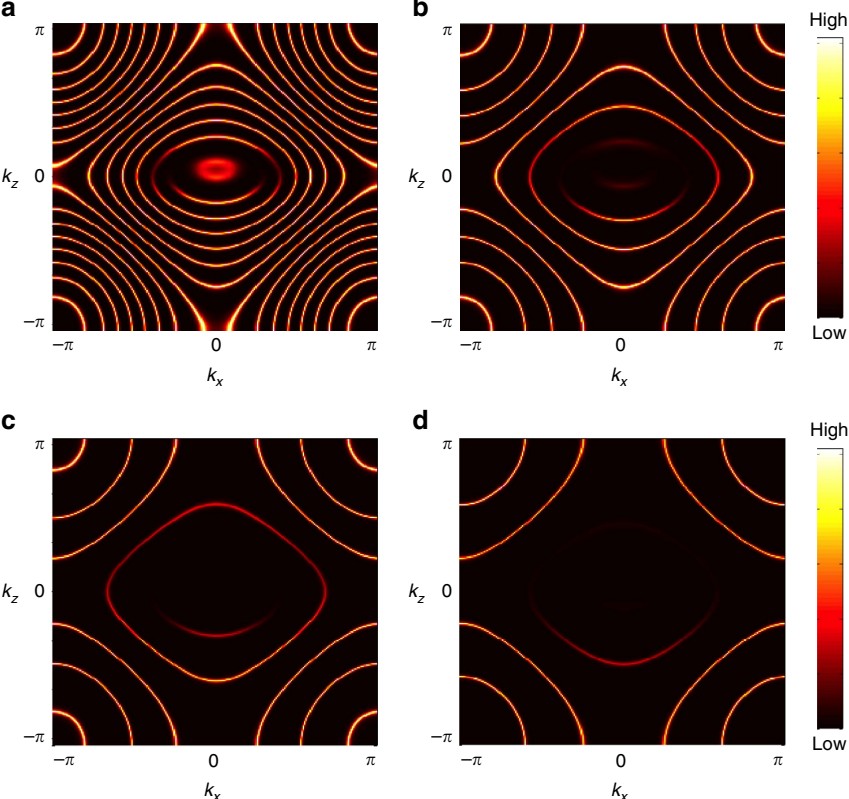

**Figure 2 | Evolution of the Wannier–Stark ladder as a function of electric-field strength.** Specifically, the electric-field strengths are set equal to $eaE/t = 0.25, 0.5, 0.75, 1$ at **a–d**, respectively. Here, WSL eigenstates are obtained by directly diagonalizing the microscopic model Hamiltonian under electric fields.

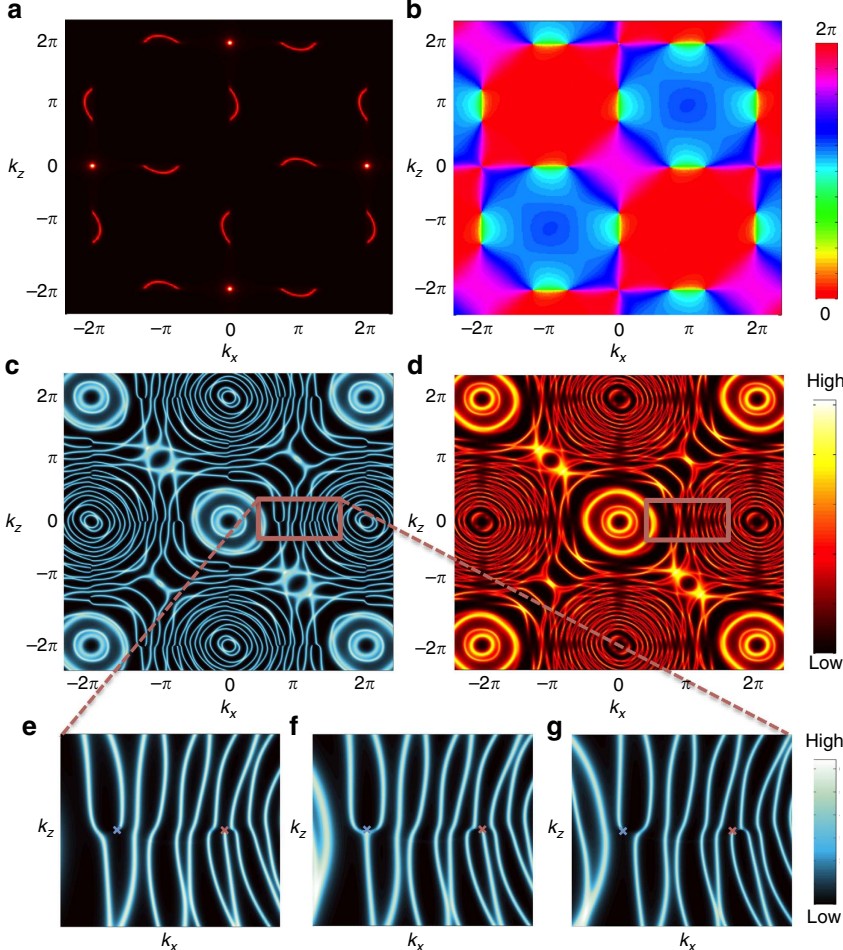

**Figure 3 | Surface and bulk Fermi arcs in a time-reversal invariant Weyl semimetal.** Here, we analyse the model Hamiltonian proposed by Ojanen in equation (6) with model parameters chosen so that $k_0 = 3\pi/4$ and $\lambda/t = 0.25$, which determine $\epsilon/t$ via $\epsilon/4\lambda = \sin k_0/2$. (**a**) Zero-energy momentum spectrum of $y$ axis-cut surface states showing the trajectories of multiple surface Fermi arcs. (**b**) Zak phase of the top valence band integrated along the $y$ axis. (**c**) Zero-energy momentum spectrum of WSL eigenstates generated from both top and bottom valence bands, which is obtained by diagonalizing the NASH in equation (8) and computing the spectral function of so-obtained WSL eigenstates. Here, the electric field is applied along the $y$ direction with its strength set equal to $eaE/t = 0.2$. (**d**) Zero-energy momentum spectrum of WSL eigenstates, which is obtained by directly diagonalizing the microscopic model Hamiltonian under an electric field with the same strength. (**e**–**g**) Magnified views of the boxed region for different energy cuts at $\omega/eaE = -0.02$, $0.02$, $0.06$, respectively. Projected Weyl nodes are marked by blue and red x's with different colours denoting different chiralities.

$(0, 2\pi, \pm k_0)$, $(2\pi, 0, \pm k_0)$ and $(2\pi, \pm k_0, 0)$, where $k_0 = 2\sin^{-1}(\epsilon/4\lambda)$. Meanwhile, $H_\sigma^+$ $(H_\sigma^-)$ describes a topologically trivial insulator composed of the outer two bands.

We check if such a time-reversal invariant Weyl semimetal can be also characterized by the existence of bulk Fermi arcs. To this end, it is important to realize that the band structure of time-reversal invariant Weyl semimetal is generally more complicated than that of the time-reversal symmetry-broken counterpart due to various band crossings. In the above Hamiltonian, it turns out that there are crossings between the top and bottom valence/conduction bands. In this situation, the WSL eigenenergy cannot be simply given by the adiabatic formula in equation (2), but rather obtained as an eigenvalue solution of the so-called non-Abelian semiclassical Hamiltonian (NASH)[23]:

$$\mathcal{H}_{\mathrm{NAS}}(\mathbf{k}) = \hat{\mathcal{E}}(\mathbf{k}) + e\mathbf{E} \cdot \left[ i\nabla_{\mathbf{k}} + \hat{\mathcal{A}}(\mathbf{k}) \right], \qquad (8)$$

where $[\hat{\mathcal{E}}]_{\alpha\beta} = \delta_{\alpha\beta}\mathcal{E}_\alpha$ is the energy dispersion and $[\hat{\mathcal{A}}]_{\alpha\beta} = \langle\phi_\alpha|i\nabla_{\mathbf{k}}|\phi_\beta\rangle$ is the Berry connection with a non-Abelian structure[27–32]. Here, $\alpha$ and $\beta$ denote the indices of all bands that cross each other. Note that the NASH eigenvalue equation

can be exactly solved by the adiabatic formula if all off-diagonal elements of the non-Abelian Berry connection are set equal to zero[23]. See Methods for details on how to diagonalize the NASH efficiently to obtain the spectral function of WSL eigenstates.

Figure 3a shows the zero-energy momentum spectrum of $y$ axis-cut surface states, which exhibits multiple surface Fermi arcs. Considering that the block-diagonalized Hamiltonian of the middle two bands can be regarded as essentially a lattice-regularized Hamiltonian of Weyl semimetal containing all Weyl nodes, it is natural to predict that the connectivity of the above surface Fermi arcs is determined by the $\pi$ Zak-phase condition, where the Zak phase is obtained by integrating the Berry connection of the top valence (or bottom conduction) band along the $y$ axis. Figure 3b shows that this prediction is indeed true with excellent accuracy.

More importantly, Fig. 3c shows that each and every projected Weyl node creates a topological defect of the WSL. Specifically, see the magnified views of the boxed region (Fig. 3e–g), which contains only two projected Weyl nodes. As one can see, there exists an edge dislocation exactly at each and every projected Weyl node. While appearing as three-way crossings at special

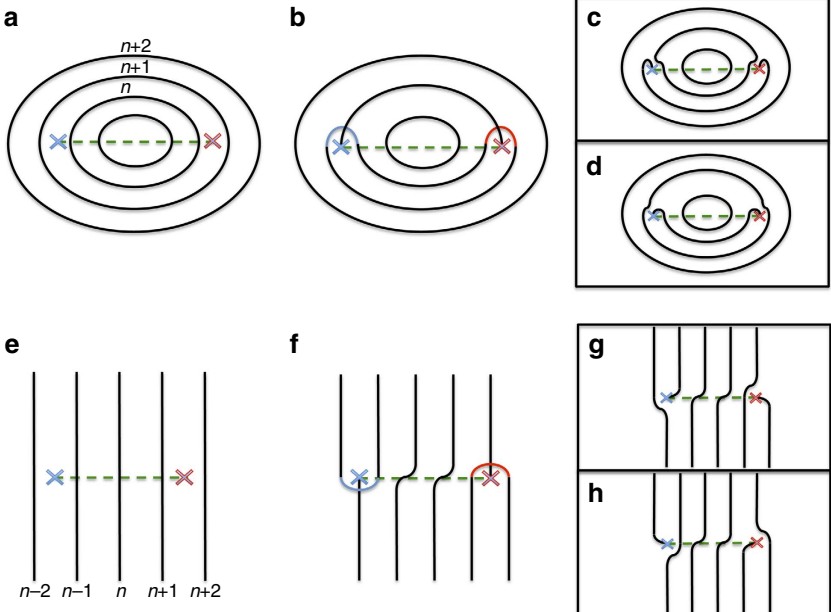

**Figure 4 | Heuristic explanation for the formation of bulk Fermi arcs.** There are two distinct situations: (i) (**a**–**d**) where WSL eigenstates form concentric layers between two projected Weyl nodes; and (ii) (**e**–**h**), where they form parallel layers. Projected Weyl nodes are marked by blue (denoting the $+1$ chirality) and red ($-1$) x's, around which the Zak phase winds by $2\pi$ when encircling counter-clockwise and clockwise, respectively. (**a**,**e**) Depiction of the situations, where the Zak phase is ignored. Dashed lines denote the $\pi$ Zak-phase curves, along which surface Fermi arcs are expected to occur. With inclusion of the Zak phase, the WSL eigenstates with two different indices $n$ and $n+1$ are smoothly fused together encircling a projected Weyl node. (**b**,**f**) Depiction of how this fusion can occur at a certain special energy cut, causing three-way crossings of WSL eigenstates. Generally, below and above such a special energy cut ((**c**)/(**g**) and (**d**)/(**h**), respectively), three-way crossings get split in such a way that open-line segments, that is, bulk Fermi arcs are formed.

energy cuts, for example, in Fig. 3f, topological defects of the WSL are generically end points of an open-line segment, which is nothing but the bulk Fermi arc. Below, we provide a heuristic explanation for the formation of these bulk Fermi arcs as well as the previous one in Fig. 1.

Before doing so, it is important to mention that the above sharp structure of topological defects gets softened in the presence of mixing between WSL eigenstates generated from all four bands including the top/bottom valence/conduction bands. Fortunately, even with this mixing, the peculiar topological structure of Weyl semimetal is still clearly visible as a misalignment of WSL eigenstates near projected Weyl nodes. See the boxed region in Fig. 3d in comparison with that in Fig. 3c.

Figure 4 provides a heuristic explanation for the formation of bulk Fermi arcs. For simplicity, we first discuss the adiabatic situation described by equation (2), assuming that the Zak phase plays a deciding role in determining the topology of WSL eigenstates. The Zak phase winds by $2\pi$ either counter-clockwise or clockwise around each projected Weyl node. This means that the WSL eigenstates with two different indices $n$ and $n+1$ can be smoothly fused together encircling a projected Weyl node. Such a fusion can cause three-way crossings of WSL eigenstates, creating topological defects of the WSL. At general energy cuts, these three-way crossings get split in such a way that bulk Fermi arcs are formed. As mentioned above, while softened, this structure of topological defects remains intact even in non-adiabatic situations, where mixings are allowed between WSL eigenstates generated from different bands.

**Weyl fermion number conservation under an electric field.** We argue that the existence of bulk Fermi arcs is actually required to conserve the Weyl fermion number under an electric field. It was shown by Nielsen and Ninomiya[13] that the chiral anomaly of

Weyl fermion can be resolved by considering Weyl fermions in a crystal, or in a lattice-regularized theory. Specifically, when parallel electric and magnetic fields are applied along the line connecting two Weyl nodes, the displacement of the Fermi surface (that is, the Weyl fermion creation/annihilation) in one Weyl node is exactly compensated by that in the other since both Fermi surfaces are interconnected below through a one-dimensional bulk conduction channel composed of filled states, therefore conserving the Weyl fermion number.

Now, let us imagine what happens when parallel electric and magnetic fields are applied perpendicular to the Weyl-node connecting line. In this situation, the Fermi surfaces of two Weyl nodes are not interconnected through a single one-dimensional bulk conduction channel. To conserve the Weyl fermion number, additional conduction channels are necessary, which are provided by none other than surface Fermi arcs. Eventually, the whole conduction process forms a closed circuit composed of two surface Fermi arcs in both sides of the surface and two one-dimensional bulk conduction channels, via which Weyl fermions can travel freely through the bulk between two surface-projected copies of each Weyl node. Note that this conduction process has been predicted to cause an intriguing quantum oscillation in Weyl semimetal[33–35].

There is, however, a hidden problem when this argument is applied to the situation with finite electric fields. Under any finite electric fields, the surface Fermi arc in one side is energetically far separated from that in the opposite (provided that the system is macroscopically large). This means that the whole conduction process cannot form a closed circuit at the same energy level. A resolution of this problem is that there exist many bulk Fermi arcs in conjunction with surface counterparts, which form a chain of many closed circuits, eventually connecting both sides of the

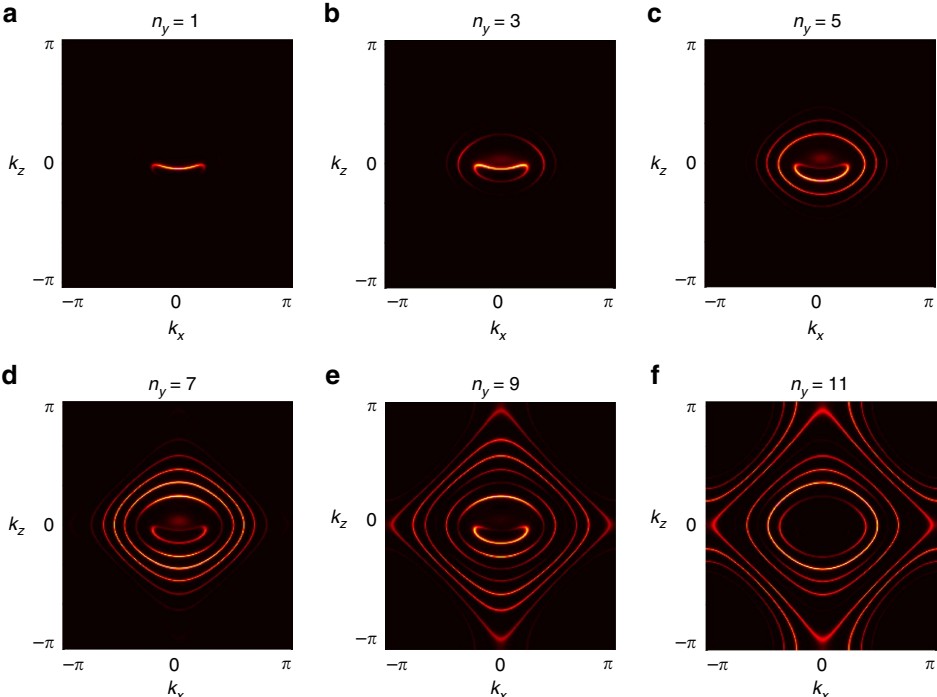

**Figure 5 | Layer-by-layer spectra of the Wannier–Stark ladder showing the evolution from a surface to a bulk Fermi arc.** Here, we analyse the model Hamiltonian in equation (3) under an electric field with $y$ axis-cut surfaces. $n_y$ denotes the layer index measured from a $y$ axis-cut surface at $n_y = 1$. The electric field is applied along the $y$ direction with its strength set equal to $eaE/t = 0.5$. Other model parameters are chosen so that $k_0 = 2\pi/7$ and $m/t = 2$. As one can see, there is a clear surface Fermi arc at $n_y = 1$ (**a**), which is slightly deformed from that in the absence of electric field. As we go into the bulk, two prominent features are observed: (i) more and more closed-loop WSL eigenstates appear surrounding the surface Fermi arc; and (ii) the surface Fermi arc is joined with a partner Fermi arc at two projected Weyl nodes, forming a closed circuit together (**b**,**c**,**d**). At a certain depth into the bulk, the surface Fermi arc disappears, while the partner Fermi arc survives (**e**). Eventually, the partner Fermi arc itself disappears deep inside the bulk (**f**). This partner Fermi arc is the first in a series of many bulk Fermi arcs forming the periodic structure of the WSL. The first bulk Fermi arc has the maximum intensity at the depth of **e** for this particular value of energy cut, which is set to maximize the intensity of the surface Fermi arc. Other bulk Fermi arcs appear at deeper layers with correspondingly higher values of energy cut.

surface. Below, we provide explicit numerical evidence supporting this argument.

Figure 5 shows layer-by-layer constant-energy momentum spectra of WSL eigenstates in the model Hamiltonian in equation (3) under an electric field with $y$ axis-cut surfaces. Specifically, we compute the following spectral function:

$$\rho_{n_y}(\omega, \mathbf{k}_\perp) = -\frac{1}{\pi} \mathrm{Im} \mathrm{Tr}' \left\langle n_y \left| \frac{1}{\omega - \tilde{H}'(\mathbf{k}_\perp) - V + i\eta} \right| n_y \right\rangle, \quad (9)$$

where $n_y$ is the layer index and the trace $\mathrm{Tr}'$ is taken over only the pseudospin index. $\tilde{H}'$ is the same as $\tilde{H}$ in equation (5) except that, here, a $y$ axis-cut surface is located at $n_y = 1$. It is important to note that the surface Fermi arc is joined with a partner Fermi arc at two projected Weyl nodes, forming a closed circuit together. This partner Fermi arc is the first in a series of many bulk Fermi arcs forming the periodic structure of the WSL.

One may ask how the connectivity of surface Fermi arcs evolves into that of bulk Fermi arcs. In the above example, the two connectivities happen to be the same, but in general can be very different, as seen in Fig. 4. As explained previously, the connectivity of surface Fermi arcs is predominantly determined by the $\pi$ Zak-phase condition, while that of bulk Fermi arcs is determined by a delicate interplay between the Zak phase and the band dispersion.

## Discussion

In this work, we have shown that Weyl nodes, which are responsible for the peculiar topological structure of Weyl semimetal, can be directly visualized as topological defects of the WSL emerging under an electric field. This opens up the possibility of a novel spectroscopic method to characterize Weyl semimetal. Below, we discuss briefly how this method can be realized in experiments.

So far, the WSL has been observed only in artificial structures such as semiconductor superlattices[36,37] and optical lattices[38] due to the fact that the lattice spacing in a natural crystal is usually too small that a strong electric field is necessary to generate sufficiently well-developed WSL spectral lines; for typical experimental situations, the necessary electric-field strength is estimated to be around the order of $100\,\mathrm{kV\,cm^{-1}}$ (ref. 23). To overcome this obstacle, there may be two possible strategies: (i) constructing a Weyl semimetal with a large lattice spacing; or (ii) applying a strong electric field without damaging the sample.

For the first strategy, it has been proposed[39,40] that a Weyl semimetal can be constructed in a superlattice system composed of alternating layers of 3D topological insulators and ordinary insulators. Meanwhile, there has been a recent outburst of various proposals for constructing Weyl semimetals in optical lattice systems with cold atoms[41–46]. Our method can be particularly useful for such cold-atom Weyl semimetals in optical lattice systems, which are known to suffer from various detection issues: (i) edges/surfaces are not well defined[47]; and (ii) transport measurements are limited, or have different characteristics from those in condensed matter systems[48]. Our method, which detects a bulk property in a non-transport measurement, could be an ideal alternative.

For the second strategy, various pump–probe techniques can be useful since a strong electric field can be applied in the form of pulse or radiation without damaging the sample[49,50].

## Methods

**Diagonalization of the non-Abelian semiclassical Hamiltonian.** Here, we discuss how to diagonalize the NASH efficiently to obtain the spectral function of WSL eigenstates. One method is to Fourier-transform the NASH from the momentum to the real space, which involves Fourier-transforming both energy dispersion $\hat{\mathcal{E}}(\mathbf{k})$ and non-Abelian Berry connection $\hat{\mathcal{A}}(\mathbf{k})$ (ref. 23). Unfortunately, this method turns out to be inefficient in Weyl semimetal due to a slow convergence of the truncation error for higher-order Fourier components.

A more efficient alternative is to rewrite the differential operator $i\nabla_{\mathbf{k}}$ in a discrete momentum representation, which is convenient for numerical diagonalization. To this end, it is important to note that $i\nabla_{\mathbf{k}}$ is in fact the position operator $\hat{\mathcal{R}}$, which is represented as a matrix in the momentum space as follows:

$$\langle \mathbf{k}'|\hat{\mathcal{R}}|\mathbf{k}''\rangle = i\nabla_{\mathbf{k}'}\delta(\mathbf{k}'-\mathbf{k}''). \tag{10}$$

From this forward, let us focus on position and momentum components parallel to the electric field, which are denoted as $\mathcal{R}_\parallel$ and $k_\parallel$, respectively.

Next, we note the following representation of the delta function by using the so-called Dirichlet kernel:

$$\delta(k_\parallel) = \lim_{N\to\infty} D_N(k_\parallel) = \lim_{N\to\infty} \frac{\sin(k_\parallel(N+1/2))}{2\pi\sin(k_\parallel/2)}, \tag{11}$$

where $k_\parallel = k_\parallel' - k_\parallel''$. Motivated by this equality, we replace the delta function by its discrete version:

$$\delta_{\text{disc}}(k_\parallel) = \frac{1}{2N+1}\frac{\sin(k_\parallel(N+1/2))}{\sin(k_\parallel/2)}, \tag{12}$$

where $k_\parallel = 2\pi j/(2N+1)$ with $j = -N, -N+1, \ldots, N-1, N$. This leads to a matrix representation of the position operator $\hat{\mathcal{R}}_\parallel$ in the discrete momentum space:

$$\left\langle k_\parallel'|\hat{\mathcal{R}}_\parallel|k_\parallel''\right\rangle = i\frac{\partial}{\partial k_\parallel'}\delta_{\text{disc}}\left(k_\parallel'-k_\parallel''\right). \tag{13}$$

This representation may seem natural in a slightly different, but more physical perspective; what we have done is basically equivalent to Fourier-transforming the position operator from the real lattice space with a finite-length $L = 2N+1$ to the discrete parallel momentum space with $k_\parallel = 2\pi j/L$ with $j = -N, -N+1, \ldots, N-1, N$.

It is important to note that $\hat{\mathcal{E}}(\mathbf{k})$ is a simple diagonal matrix with respect to both band and discrete parallel momentum indices. On the other hand, $\hat{\mathcal{A}}(\mathbf{k})$ is a $2\times 2$ matrix with generally non-zero off-diagonal elements with respect to the band index, while being an $N\times N$ diagonal matrix with respect to the discrete parallel momentum index. Of course, $\hat{\mathcal{R}}_\parallel$ is a diagonal matrix with respect to the band index. With the knowledge of all these operators in the above discrete momentum representation, the NASH can be diagonalized to generate WSL eigenstates as a function of perpendicular momentum, $\mathbf{k}_\perp$. Specifically, we compute the following spectral function of WSL eigenstates obtained from the NASH:

$$\rho(\omega, \mathbf{k}_\perp) = -\frac{1}{\pi}\text{ImTr}\left[\frac{1}{\omega - \mathcal{H}_{\text{NAS}}(\mathbf{k}) + i\eta}\right], \tag{14}$$

where the trace Tr is taken over both band and discrete parallel momentum indices.

**Data availability.** All relevant data as well as computer codes are available from the authors.

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

## Acknowledgements

We are grateful to Jainendra K. Jain, Kai Sun, Naoto Nagaosa, Takahiro Morimoto, Gil Young Cho, Jung Hoon Han, Hong Yao and Hyun-Woong Kwon for insightful discussions. This work is partially supported by the NSERC of Canada and Canadian Institute for Advanced Research (Y.B.K.).

## Author contributions

K.W.K. performed all the relevant analytical as well as numerical calculations under the supervision of K.P. The initial idea was conceived by K.W.K. based on the work by W.-R.L. and K.P. who provided details on the analysis techniques. Y.B.K. made various contributions to the interpretation of the results. K.P. wrote the manuscript.

## Additional information

**Competing financial interests:** The authors declare no competing financial interests.

