## [Peer Review File · Nature Communications]

REVIEWERS' COMMENTS:

Reviewer #1 (Remarks to the Author):

The authors propose a theory explaining the connectivity of the Fermi arcs in Weyl semimetals. In my opinion the novelty and the effectiveness of the approach are not illustrated in a solid way. For example, a Berry (Zak phase of π necessarily leads to the appearance of a metallic end state in 1d in inversion-symmetric systems (Vanderbilt-King-Smith PRB'93)

The authors also cite the PRB of Mong, which also contains a bunch of relevant information. It is necessary to explain what is new in the present approach, and how it can be experimentally tested. All the provided illustrations are based on models, none is given for a real material example, while now there are many.

Reviewer #2 (Remarks to the Author):

This paper asks what determines the connectivity of surface Fermi arcs in, say, inversion symmetry breaking Weyl semimetals, where multiple pairs of Weyl nodes are present. To answer this question, the authors have shown that the locations of surface Fermi arcs are predominantly determined by the condition that the Zak phase integrated along the normal direction to the surface is π . This idea is an extension of the original similar idea applied to graphene. More importantly, the Zak phase in WSM reveals the peculiar topological structure of Weyl semimetal directly in the bulk. The authors have shown that the non-trivial winding of the Zak phase around each projected Weyl node manifests itself as a topological defect of the Wannier-Stark ladder, the energy eigenstates emerging under an electric field. Remarkably, this structure leads to "bulk Fermi arcs," i.e., open line segments in the bulk momentum spectra.

I find the paper to be original and of interest to the growing community of topological order in condensed matter systems. The conclusions seem to be robust, valid, and reliable. The writing and presentation of the paper are also reasonably good. I only wish that, for the sake of completeness, the authors discussed the concept of WSL in a little bit more detail. When that is done, I recommend the paper to be published in Nature Communication.

Reviewer #3 (Remarks to the Author):

In this manuscript the authors showed that the locations of surface Fermi arcs in a Weyl semimetal are predominantly determined by the condition that the Zak phase integrated along the normal direction to the surface is π . Then they showed that the non-trivial winding of the Zak phase around each projected Weyl node manifests itself as a topological defect of the Wannier-Stark ladder, the energy eigenstates emerging under an electric field. They argued that bulk Fermi arcs should exist in conjunction with the surface counterparts to conserve the Weyl fermion number under an electric field, which was supported by explicit numerical evidence.

Weyl semimetal has emerged as an active research topic in condensed matter. One can easily find articles on this topic from high impact journals such as Nature Communication. Among various important yet unanswered issues, the question that "what determines the connectivity of surface Fermi arcs when multiple pairs of Weyl nodes are present" is of general interest. Therefore, their answer to this question is a very important result and pointed out a possible spectroscopic method to

characterize Weyl semimetal. To the best of my knowledge, they are the first group who gave the answer. This approach could be tried to cold atom Weyl semimetal in optical lattice systems. This work provides useful information to people working in the same field, and may give hints to people working in similar fields.

Computationally, the method used by the authors to diagonalize the non-Abelian semi-classical Hamiltonian and obtain the spectral function is reliable and efficient. I went through it briefly and did not find anything seeming wrong. The method may also apply to similar Hamiltonian so readers may learn something here. Physical arguments, mathematical derivations, and figures in this manuscript are clear. The analyses presented are convincing.

References are adequate. Thus the achievement in this work is nontrivial and I recommend publication of this manuscript.

Response to the report of Reviewer #1

Reviewer #1 made the following comments in his/her report.

“The authors propose a theory explaining the connectivity of the Fermi arcs in Weyl semimetals. In my opinion the novelty and the effectiveness of the approach are not illustrated in a solid way. For example, a Berry (Zak phase of π) necessarily leads to the appearance of a metallic end state in 1d in inversion-symmetric systems (Vanderbilt-King-Smith PRB’93)

The authors also cite the PRB of Mong, which also contains a bunch of relevant information. It is necessary to explain what is new in the present approach, and how it can be experimentally tested. All the provided illustrations are based on models, none is given for a real material example, while now there are many.”

The above report of Reviewer #1 is composed of the following, two points, to which we respond as follows.

- (1) “In my opinion the novelty and the effectiveness of the approach are not illustrated in a solid way. For example, a Berry (Zak phase of π) necessarily leads to the appearance of a metallic end state in 1d in inversion-symmetric systems (Vanderbilt-King-Smith PRB’93) The authors also cite the PRB of Mong, which also contains a bunch of relevant information. It is necessary to explain what is new in the present approach, and how it can be experimentally tested.”

We are sorry to find that Reviewer #1 did not fully appreciate what is new in our approach. Unfortunately, we think that the above comments are rather odd, judging from what Reviewer #1 mentioned in regard to the following two references, (i) King-Smith and Vanderbilt, PRB 47, 1651 (1993) and (ii) Mong and Shivamoggi, PRB 83, 125109 (2011).

First, as mentioned by Reviewer #1, the paper by King-Smith and Vanderbilt discusses 1D systems with the inversion symmetry, in which case the Zak phase can have the value of either 0 or π only. Meanwhile, in our paper, we discuss 3D Weyl-semimetal systems with/without the inversion symmetry, where the Zak phase can have any values between 0 and 2π . While the general concept of the Zak phase is the same, the situations are completely different. In particular, it is made clear in our work that the Zak phase has a non-trivial topological winding, or screw dislocation

around each projected Weyl node, which is completely new. This point has been emphasized numerous times throughout our manuscript. It is odd that Reviewer #1 did not get this point. We would like to emphasize that we are the first group to make the connection between this non-trivial topological structure of the Zak phase and the surface Fermi arc, as pointed out by another referee, Reviewer #3.

(We do not think that the paper by King-Smith and Vanderbilt has a direct relevance to the main results of our work. However, it would be nice to cite this paper in an appropriate place. Thus, we have revised our manuscript to add this paper as a reference.)

Also, while providing “a bunch of relevant information,” Mong and Shivamoggi did not realize in their paper that there is a close connection between their geometrical formula (which relates the existence of surface states to properties of the bulk Hamiltonian) and the Zak phase itself. Without knowing this connection, it is quite difficult to construct the general mathematical condition for the locations of surface Fermi arcs, as done in our work.

Finally, Reviewer #1 mentioned that it is necessary to explain how the new results obtained in our paper can be experimentally tested. This comment is not only strange, but also quite misleading since we have in fact discussed the experimental realization of our results in the Discussion section in great details. We do not understand how Reviewer #1 was able to miss this.

(2) “All the provided illustrations are based on models, none is given for a real material example, while now there are many.”

There are several reasons that we have focused on the model Hamiltonians in this work.

First, by nature, topological properties are robust against various other material-specific properties. Therefore, as long as we are interested in the topological properties, we can focus on the model Hamiltonians that can capture the correct topological properties with the right symmetry of interest such as the time reversal and/or inversion symmetries. This is exactly what was done in our work.

Second, this is the work, where we have proposed our idea for the first time. Therefore, it is necessary to firmly establish the validity of all the detailed analytical as well as numerical techniques before using them in more realistic situations. Perhaps, we can extend our approach to real materials in the future.

Third, most importantly, being the subject of main interest in this work, the Wannier-Stark ladder is obtained in the presence of electric field, which necessarily breaks the translational symmetry. Unfortunately, the density functional theory, which is the best theoretical tool for analyzing real materials, cannot describe such a situation very well since it requires the periodic structure of the lattice. Of course, a more realistic tight-binding model can be constructed from the information obtained from the density functional theory. However, we would like to emphasize that, at the end of the day, one must resort to model Hamiltonians, as done in our work, in order to treat the effects of electric field accurately.

Response to the report of Reviewer #2

Reviewer #2 made the following comments in his/her report.

“This paper asks what determines the connectivity of surface Fermi arcs in, say, inversion symmetry breaking Weyl semimetals, where multiple pairs of Weyl nodes are present. To answer this question, the authors have shown that the locations of surface Fermi arcs are predominantly determined by the condition that the Zak phase integrated along the normal direction to the surface is π . This idea is an extension of the original similar idea applied to graphene. More importantly, the Zak phase in WSM reveals the peculiar topological structure of Weyl semimetal directly in the bulk. The authors have shown that the non-trivial winding of the Zak phase around each projected Weyl node manifests itself as a topological defect of the Wannier-Stark ladder, the energy eigenstates emerging under an electric field. Remarkably, this structure leads to “bulk Fermi arcs,” i.e., open line segments in the bulk momentum spectra.

I find the paper to be original and of interest to the growing community of topological order in condensed matter systems. The conclusions seem to be robust, valid, and reliable. The writing and presentation of the paper are also reasonably good. I only wish that, for the sake of completeness, the authors discussed the concept of WSL in a little bit more detail. When that is done, I recommend the paper to be published in Nature Communication.”

After summarizing the main results of our paper, Reviewer #2 recommended our paper to be published in Nature Communications once we provide a little bit more detailed discussion on the concept of Wannier-Stark ladder (WSL).

In response to the referee’s suggestion, we have revised our manuscript to add detailed discussions in the following two places: (i) the third paragraph in the Introduction section and (ii) the first and second paragraphs in the subsection titled “Topological defects of the Wannier-Stark ladder” in the Results section.

Response to the report of Reviewer #3

Reviewer #3 made the following comments in his/her report.

“In this manuscript the authors showed that the locations of surface Fermi arcs in a Weyl semimetal are predominantly determined by the condition that the Zak phase integrated along the normal direction to the surface is π . Then they showed that the non-trivial winding of the Zak phase around each projected Weyl node manifests itself as a topological defect of the Wannier-Stark ladder, the energy eigenstates emerging under an electric field. They argued that bulk Fermi arcs should exist in conjunction with the surface counterparts to conserve the Weyl fermion number under

an electric field, which was supported by explicit numerical evidence.

Weyl semimetal has emerged as an active research topic in condensed matter. One can easily find articles on this topic from high impact journals such as Nature Communication. Among various important yet unanswered issues, the question that “what determines the connectivity of surface Fermi arcs when multiple pairs of Weyl nodes are present” is of general interest. Therefore, their answer to this question is a very important result and pointed out a possible spectroscopic method to characterize Weyl semimetal. To the best of my knowledge, they are the first group who gave the answer. This approach could be tried to cold atom Weyl semimetal in optical lattice systems. This work provides useful information to people working in the same field, and may give hints to people working in similar fields.

Computationally, the method used by the authors to diagonalize the non-Abelian semi-classical Hamiltonian and obtain the spectral function is reliable and efficient. I went through it briefly and did not find anything seeming wrong. The method may also apply to similar Hamiltonian so readers may learn something here. Physical arguments, mathematical derivations, and figures in this manuscript are clear. The analyses presented are convincing. References are adequate. Thus the achievement in this work is nontrivial and I recommend publication of this manuscript.”

As one see from the report, Reviewer #3 recommended publication of our manuscript without raising any issues.